# Oligometastatic Disease (OMD): The Classification and Practical Review of Prospective Trials

**DOI:** 10.3390/cancers15215234

**Published:** 2023-10-31

**Authors:** Timur Izmailov, Sergey Ryzhkin, Gleb Borshchev, Sergei Boichuk

**Affiliations:** 1Pirogov National Medical and Surgical Center, Ministry of Health of Russia, Moscow 127994, Russia; izmaylovtr@pirogov-center.ru (T.I.); borshchevgg@pirogov-center.ru (G.B.); 2Department of Radiotherapy and Radiology, Russian Medical Academy of Continuous Professional Education, Moscow 127051, Russia; ryzhkinsa@rmapo.ru; 3Department of Hygiene, Kazan State Medical University, Kazan 420012, Russia; 4Department of Pathology, Kazan State Medical University, Kazan 420012, Russia; 5“Biomarker” Research Laboratory, Institute of Fundamental Medicine and Biology, Kazan Federal University, Kazan 420008, Russia

**Keywords:** oligometastatic disease (OMD), cancer, oligometastases, systemic therapy, breast cancer (BC), non-small cell lung cancer (NSCLC), colorectal cancer (CRC), prostate cancer (PC), progression-free survival (PFS), overall survival (OS), stereotactic body radiotherapy (SBRT)

## Abstract

**Simple Summary:**

Oligometastatic disease (OMD) is currently recognized as an intermediate state of cancer between the localized and widely metastatic form of the disease. Before the term OMD was introduced into practical oncology, metastatic lesions were considered markers of the systemic disease and the majority of such patients received systemic therapies that were usually associated with a broad spectrum of adverse side effects. After the concept of the OMD state of cancer was introduced into the clinic and accepted by oncologists, different treatment options became available. Given no OMD biomarkers are currently available, the diagnosis of OMD is based exclusively on imaging findings. Based on this data, different clinical scenarios have become available that are associated with different treatment strategies and clinical outcomes. Some of clinical studies illustrated a significant improvement in the overall survival (OS) and progression-free survival (PFS) rates of cancer patients who received local aggressive therapies (e.g., stereotactic body radiotherapy (SBRT)). Such therapies are generally well-tolerated and provide good local control of the disease. Moreover, local therapies can be used to delay systemic therapies in the context of metachronous disease. We discuss here the current classification and management strategies in OMD and review the prospective and ongoing clinical trials.

**Abstract:**

Oligometastatic disease (OMD) is currently known as an intermediate state of cancer, characterized by a limited number of systemic metastatic lesions for which local ablative therapy could be curative. Indeed, data from multiple clinical trials have illustrated an increase in overall survival (OS) for cancer patients when local ablative therapy was included in the systemic adjuvant therapy. Given that no driver and somatic mutations specific to OMD are currently established, the diagnosis of OMD is mainly based on the results of X-ray studies. In 2020, 20 international experts from the European Society for Radiotherapy and Oncology (ESTRO) and the European Organization for Research and Treatment of Cancer (EORTC) developed a comprehensive system for the characterization and classification of OMD. They identified 17 OMD characteristics that needed to be assessed in all patients who underwent radical local treatment. These characteristics reflect the tumor biology and clinical features of the disease underlying the development of OMD independently of the primary tumor type and the number of metastatic lesions. In particular, the system involves the characteristics of the primary tumor (e.g., localization, histology, TNM stage, mutational status, specific tumor markers), clinical parameters (e.g., disease-free interval, treatment-free interval), therapies (e.g., local, radical or palliative treatment, the numbers of the therapeutic regimens), and type of OMD (e.g., invasive). Based on the aforementioned criteria, an algorithm was introduced into the clinic to classify OMDs collectively according to their nomenclature. A history of polymetastatic disease (PMD) prior to OMD is used as a criterion to delineate between induced OMD (previous history of PMD after successful therapy) and genuine OMD (no history of PMD). Genuine OMD is divided into two states: recurrent OMD (i.e., after a previous history of OMD) and de novo OMD (i.e., a first newly diagnosed oligometastatic disease). de novo OMD is differentiated into synchronous and metachronous forms depending on the length of time from the primary diagnosis to the first evidence of OMD. In the case of synchronous OMD, this period is less than 6 months. Lastly, metachronous and induced OMD are divided into oligorecurrence, oligoprogression, and oligopersistence, depending on whether OMD is firstly diagnosed during an absence (oligo recurrence) or presence (oligoprogression or oligopersistence) of active systemic therapy. This classification and nomenclature of OMD are evaluated prospectively in the OligoCare study. In this article, we present a practical review of the current concept of OMD and discuss the available prospective clinical trials and potential future directions.

## 1. Introduction

Despite being an intermediate state between localized cancer and the metastatic form of the disease that was initially named the “oligometastatic state” almost 25 years ago, debates regarding the fundamental mechanisms, therapeutic strategies, and subgroups of the oligometastatic disease (OMD) are still ongoing. The results of the local treatment modalities (surgery and/or radiotherapy) of OMD clearly illustrate the possibility of curing patients with distinct forms of cancer [1,2,3,4]. Despite the fact that the introduction of novel technologies in cancer radiotherapy significantly increased progression-free survival (PFS) and overall survival (OS) rates for patients with a broad spectrum of malignancies, the benefits of radiotherapy for patients with OMD are still debatable [5,6,7].

Based on the evidence of multistep cancer progression, *Hellman* and *Weichselbaum* originally described the oligometastatic state in 1995. They proposed the existence of an oligometastatic state as an intermediate state between localized and widely metastatic lesions [8]. The time interval between this initial publication and subsequent publications describing the oligometastatic state was more than 10 years. The vast majority of publications represent retrospective studies, including either single (n = 50) or multicenter (n = 23) trials. Some of the publications are based on the data obtained from prospective studies, whereas others are illustrating the results from phases I and II of clinical trials. In particular, when cancer patients with metastatic disease were analyzed in randomized phase II clinical trials, some of the trials reported significant improvements in PFS and OS rates in cancer patients when standard systemic anti-cancer therapy was supplemented with local ablative therapy of OMD [9,10,11].

In 2019, Palma D. et al. demonstrated the benefits of stereotactic body radiotherapy (SBRT) for distant metastasis supplemented with the standard therapies for patients with distinct forms of cancer, including breast cancer (BC), non-small cell lung cancer (NSCLC), colorectal cancer (CRC), and prostate cancer (PC) [12].

Ost P. et al. conducted a study on patients with an oligorecurrent form of PC, in which the efficacy of the SBRT of the metastatic lesions was compared with a group of non-treated patients. Of note, all the patients enrolled in this study were not receiving systemic androgen deprivation therapy (ADT) as an initial treatment strategy, but this therapeutic option was introduced only after the disease progression. The primary end-point of the phase II trial was ADT-free survival. Patients enrolled in this trial were randomly assigned (1:1) to either surveillance or metastasis-directed therapy (MDT) of the detected lesions (surgery or SBRT). Surveillance was based on the evaluation of prostate-specific antigen (PSA) levels, measured every 3 months, with repeated imaging procedures at PSA increase or clinical signs of disease progression. The assignment was based on two criteria: the doubling time of PSA levels (≤3 v >3 months) and identification of nodal versus non-nodal metastatic lesions. As expected, the median ADT-free survival rates were up to 9 months longer in the MDT group compared with the surveillance group (21 and 13 months, respectively) [13].

The randomized phase III STAMPEDE trial aimed to examine the impact of radiotherapy on primary metastatic PC. The primary outcome was OS, whereas secondary outcomes included failure-free survival, PFS, metastatic progression-free survival, prostate cancer-specific survival, and symptomatic local event-free survival. The authors found that radiotherapy improved failure-free survival, but not OS. Even though this trial was not focused on oligometastatic PC, the patients exhibiting a low metastatic burden demonstrated the clear benefits of this therapeutic approach. This in turn raised the possibility that local radiotherapy of primary small-volume metastatic tumors might be beneficial for patients with other malignant diseases [14].

Conversely, very little is currently known about the molecular basis of OMD (i.e., the specific features and markers of OMD preventing wide spreading of the tumors and progression of the malignancies) [15,16]. Some authors reported changes in the microRNA (miRNA) profiles that correlate with the progression of oligometastatic and polymetastatic lung cancer. In particular, the list of miRNAs in patients with oligometastatic lung cancer that are associated with low, intermediate, and high risk of progression (LRP, IRP, and HRP, respectively) included 37 miRNAs that were significantly downregulated in HRP samples [17]. Similarly, an integrated specific molecular subtype to determine the oligometastatic state in CRC with liver metastases (CRCLM) was described in detail by Pitroda S. et al. They identified three distinct molecular CRCLM subtypes that exhibited heterogeneous clinical outcomes with a 10-year OS of 37%, 64%, and 20%, respectively [18]. In contrast, Dhondt B. et al. failed to demonstrate the serum-derived miRNA biomarkers allowing for the prospective discrimination of oligo- and polymetastatic PC patients [19].

Based on the absence of specific biomarkers reflecting OMD, diagnostic imaging procedures, including multispiral computer tomography (MSCT), are currently considered the most relevant diagnostic approaches for identifying and detecting the spread of metastasis and are thereby used to differentiate between OMD and PMD to make a decision about local ablative therapy [20,21]. For example, some studies have shown that fluoro-deoxyglucose-based positron emission tomography (PET) is helpful to identify a subset of patients with NSCLC and CRC exhibiting low tumor activity who are therefore eligible for radical therapy of their local metastatic lesions [22,23]. Thus, EORTC has identified the pivotal role of diagnostic imaging procedures to standardize and optimize the clinical diagnosis of OMD. These recommendations have been published as practice guidelines [24,25]. In 2018, the EORTC guidelines for imaging were also introduced into the European Society for Medical Oncology (ESMO) clinical practice guidelines [26]. In current clinical trials, a maximum of 3–5 metastases are accepted in the context of OMD [5].

Based on the current findings on oligometastatic NSCLC, the 5-year OS rates can be variable depending of the stage of NSCLC (i.e., 86% and 8.3% in stage I and IV, respectively) [27,28,29,30].

Based on these findings, the scoring scales were introduced to determine OS rates; however, the majority of these studies were retrospective, and contained well-known limitations and errors [31,32,33]. In particular, the groups of patients were non-comparable since one group was composed of patients who received specialized SBRT in combination with systemic therapy, whereas the other groups of patients with oligometastases received stereotactic body radiotherapy (SBRT) alone without chemotherapy [31]. Despite this study highlighting SBRT’s efficacy and ability to restrain the number of metastatic lesions in patients with oligometastases, the significance of SBRT has yet to be proven. Several studies have studied patients undergoing liver surgery for colorectal cancer metastases (CRC). They are all disparate studies, but the authors justified the resection of liver metastases in patients with resectable metastases, citing high survival rates [10].

Hong J et al. published the results of randomized multi-center clinical trials illustrating the efficacy of ablative radiotherapy in patients with oligometastases. A total of 361 patients were enrolled in this study. Primary tumors included non-small cell lung (17%), colorectal (19%), and breast cancer (16%). For patients with other types of primary tumors, a long recurrence-free interval (75 months before the development of metastases) was used as an indicator of favorable prognosis. However, the cohort of these patients was limited to 14 patients only and therefore it should only be considered as a preliminary study. Despite the promising results after SBRT, many cancer patients progressed fairly quickly, therefore highlighting the necessity of better identifying the patients who are eligible for SBRT and achieve the most benefits from this type of therapy [33].

In 2020, 20 international experts from ESTRO and EORTC developed a comprehensive system for the characterization and classification of OMD. As a result, the typical characteristics and features of OMD were identified and further recommended to be assessed in all patients who received radical local treatment for OMD. This classification system turned out to be universal, was also based on the main prognostic factors of the patient and the disease, and did not require additional diagnostic testing. The classification system also reflects the fundamental biological and clinical processes that underlie the development of OMD and does not depend on the origin of the primary tumor. These OligoCare criteria will be discussed in detail below.

The OligoCare project is based on data from the ESTRO and EORTC study (EORTC 1822, EORTC-ESTRO Joint Radiation Infrastructure Group for Europe—E2RADIatE, EORTC 1811, NCT03818503), which aimed to identify the features of the patient, tumor type, stage of the disease, and therapies, that affect the OS of patients who received radiation therapy for local metastases in OMD. The eligibility criteria in OligoCare were broad enough to reflect the diversity of routine clinical practice and to highlight the relevant prognostic factors.

To provide a systematic review and meta-analysis of the OligoCare recommendations, PRISMA [34] and the Delphi technique [35] were utilized to reach a consensus regarding the typical features of OMD. As a result of the analysis of 806 publications, 26 manuscripts were further selected that met the inclusion criteria and reflected the OS and PFS in prospective studies of OMD. Based on this analysis, 10 major and 7 additional characteristics of OMD were identified, as shown in Table 1 and Figure 1.

Based on the characteristics shown in Table 1, OMD can be classified as shown in Figure 1.

In particular, to classify OMD, five general points should be addressed.

Whether the patient has a history of PMD prior to OMD being diagnosed.Whether the patient has a history of OMD prior to the current diagnosis of OMD.Whether the OMD was diagnosed for the first time 6 months after primary cancer was diagnosed.Whether OMD was diagnosed during active systemic therapy.Whether OMD is progressing on current imaging.

### 1.1. Therapeutic Strategies of OMD

It was generally accepted for a long time that local treatment of metastatic lesions, regardless of the oligometastatic or polymetastatic state, was carried out exclusively for palliative purposes. After recognizing the concept of the OMD state, the goals and therapeutic strategies for cancer patients were changed dramatically. Targeted treatment of OMD is generally aimed at prolonging the time period until the next salvage treatment, at extending the survival rates, or curing the patients. Indeed, it is well-known that the most common endpoints assessed in clinical trials include the following criteria: an increase in OS and PFS rates, withdrawal from drug therapy (e.g., androgen deprivation therapy for oligometastatic PC), an improvement of the quality of life, etc. [36,37].

Given that local therapy is primarily aimed at eliminating all of the oligometastases and potentially uncontrolled primary tumors, the achievement of the parameters indicated above and the choice of the systemic therapy will depend on the OMD state diagnosed in each particular patient.

In the case of a first newly diagnosed OMD (i.e., de novo OMD) and/or recurrent OMD, radical treatment is primarily used to cure the patient with cancer. The results of the vast majority of randomized clinical trials (except one) were based on systemic therapy as the main therapeutic approach for patients with OMD [9,10,11,13,29,38].

In the case of OMD progressing during an active primary therapy (i.e., metachronous oligoprogression) the choice for the optimal systemic drug therapy currently remains unclear and currently consists of continuing that primary therapy or switching to other drug-based protocols.

In the case of synchronous OMD, the choice for the optimal therapy is more difficult and is currently based on combined local therapy of the primary tumor and oligometastases supplemented with systemic therapy as well. On the other hand, local therapy of de novo OMD and recurrent OMD can be extended until systemic drug therapy for the PMD will be required, thus maintaining the patient’s quality of life. The effectiveness of this approach was illustrated by Ost P et al. who documented a significant delay in the initiation of androgen deprivation systemic therapy for patients with oligo-recurrent PC who received the local therapy of metastatic lesions [13].

The goals and therapeutic strategies for patients with induced OMD might vary significantly due to the development of this state in patients with polymetastatic disease which has progressed under the partial effectiveness of systemic treatment. In this regard, local treatment of induced OMD complements the systemic therapy, and not vice versa, as was shown in the case of genuine OMD. Based on the current data, most of the patients with induced OMD currently cannot be cured. For the patients with so-called induced oligorelapse, radical local treatment aims to achieve a partial or complete response. Of note, the effectiveness of local treatment might be potentiated by introducing systemic drug therapy. On the other hand, the main goal of local treatment is considered to be to prolong the interval without the use of systemic drug therapy. In some patients with induced oligoprogression, radical local treatment of oligometastasis is aimed to restore sensitivity to systemic therapy via affecting (i.e., eliminating) the oligometastases that are resistant to the current chemotherapeutic regimens. For cancer patients with induced oligopersistence, the goal of radical local treatment is mainly to achieve a partial response to systemic drug therapy. For both the groups of patients with induced oligoprogression and induced oligopersistence, therapeutic options include the continuation of systemic drug therapy and/or switching to a new line therapy.

Further therapeutic strategies will depend on multiple factors, including the effectiveness and duration of the response, prevalence, and dynamics of disease progression, the presence of concomitant diseases, tolerability of drug treatment, etc.

### 1.2. The Dynamic Model of the Oligometastatic State

It is obvious that the OMD state may change many times in each particular patient with cancer due to the disease progression, and as a result of radio- and systemic therapy, as well. An example of the dynamic assessment of the OMD state for patients with PC was described in detail by the OligoCare scientists [39].

This model illustrates the development of the multiple and distinct variants of OMD during the whole length of the patient’s disease. It might be initially diagnosed as de novo OMD or as induced OMD as part of a polymetastatic disease. Recurrent OMD might be a result of the therapy failure for the patients initially diagnosed with de novo OMD, whereas induced OMD might be developed after the failure of systemic therapy for the patients with de novo and recurrent OMDs. Patients with recurrent or induced OMD might develop dynamic transitions between oligorecurrent, oligoprogressive, and oligopersistent variants of the disease, depending on their responses to the local and systemic therapies. A newly diagnosed OMD might represent de novo OMD or might be a result of systemic therapy failure in cancer patients with polymetastatic disease (i.e., induced OMD). Recurrent OMD may be a consequence of therapy failure for OMD associated with the development of a limited number of metastatic lesions. Development of the polymetastatic form of the disease might be a result of the unsuccessful combined therapies for OMD, composed of local ablative and systemic drug-based therapies. In this particular scenario, initiation of the combined therapy leads to the development of the induced OMD.

Patients with recurrent or induced OMDs might exhibit dynamic transitions between oligo-recurrent, oligo-progressive, and oligo-persistent forms of the disease, depending on their responses to the local and systemic drug therapies. As shown in Figure 2, cancer patients with the three upper types of OMD (e.g., de novo, recurrent, and induced) might undergo a unidirectional transition to any of the subsequent states of OMD. However, this transition is not necessarily associated with the disease progression.

The importance of therapeutic intervention in recurrent OMDs is also shown in a study illustrating the decreased incidence of the formation of novel metastatic lesions and the decreased likelihood of disease progression for the patients who were diagnosed with recurrent OMD and received several cycles of SBRT [40]. Moreover, the highest survival rates in groups of patients with induced OMD who received SBRT might be also a result of the combined therapies involving targeted-based therapies for a subset of patients with NSCLC with specific driver mutations or immunotherapies for patients with melanoma.

Clinical data support this dynamic model of the oligometastatic state after several consecutive courses of radiotherapy for OMD. Klement R et al. observed that 6.6% of patients with oligometastatic NSCLC received 1–4 courses of SBRT for metastasis [41]. Similar results were described for patients with PC [42].

Thus, the characteristics and typical features of OMD indicated above should be evaluated in all groups of patients who received local therapy for OMD. Furthermore, the assessment of the dynamic model of the oligometastatic state will be helpful to improve the therapeutic strategies for cancer patients with a broad spectrum of human malignancies.

## 2. Conclusions

Collectively, novel and safe radiotherapeutic modalities, including SBRT for therapy for cancer patients with OMD, have been widely introduced into the clinical practice over the past decade. In the meantime, a significant variation was found regarding the characteristics of patients enrolled in these studies, the endpoints that were hoped to be achieved, and the definitions used in these clinical studies. These were the main driving forces to develop and establish the typical characteristics of OMD, which will be further introduced into the clinical practice to improve the clinical outcomes for the patients with OMD.

Given the lack of specific biomarkers for OMD, its status is currently defined based on clinical data and imaging results. In order to unify the diagnostic requirements, the EORTC diagnostic group introduced the minimum criteria for diagnostic imaging of OMD [25]. To develop the universal characteristics of OMD and for its further use in clinical practice, ESTRO and EORTC also initiated the OligoCare study shown on the E2-RADIatE platform (EORTC-ESTRO Radiation Infrastructure for Europe, NCT03818503). This was an international and prospective study aimed at identifying the patients’ characteristics, tumor, stage, and treatment factors that affect the OS of patients who received radical radiotherapy for OMD. The inclusion criteria used in this study were broad enough and reflected the diversity of the clinical samples and relevant prognostic factors. To overcome these issues, the classification of the various types of OMD was further developed [39].

Despite the fact that no data from randomized phase III trials are currently available, the results of clinical trials in phase II demonstrated an increase in OS for cancer patients with OMD who received SBRT [9,11,12]. Following the success of recently completed phase II trials demonstrating important clinical outcomes after SBRT, phase III clinical trials were initiated and are currently ongoing (as reviewed in [43]). In conclusion, more data are currently needed to determine the cohorts of patients with OMD who are eligible for SBRT and might receive clinical benefits from local ablative therapies. Synchronous and metachronous OMD most likely represent distinct diseases. In contrast, the other forms of OMD (e.g., oligorecurrence, oligoprogression, and oligopersistence) might be a result of genuine OMD.

Collectively, the safety and efficacy of SBRT in cancer patients with OMD are increasingly being evaluated within prospective clinical trials. However, heterogeneity in the interpretation and classification of OMD among different scientific societies has produced different inclusion criteria for the prospective trials. Despite the inclusion of targeted-based, immunotherapeutic, and chemotherapeutic modalities into the anticancer therapeutic regimes, which have improved the clinical outcomes, these approaches have inhibited the direct assessment of the efficiency of SBRT in cancer patients with OMD [43]. The currently ongoing trials are expected to provide more detailed information and clear evidence regarding the optimal treatment strategies for patients with OMD.

## Figures and Tables

**Figure 1 cancers-15-05234-f001:**
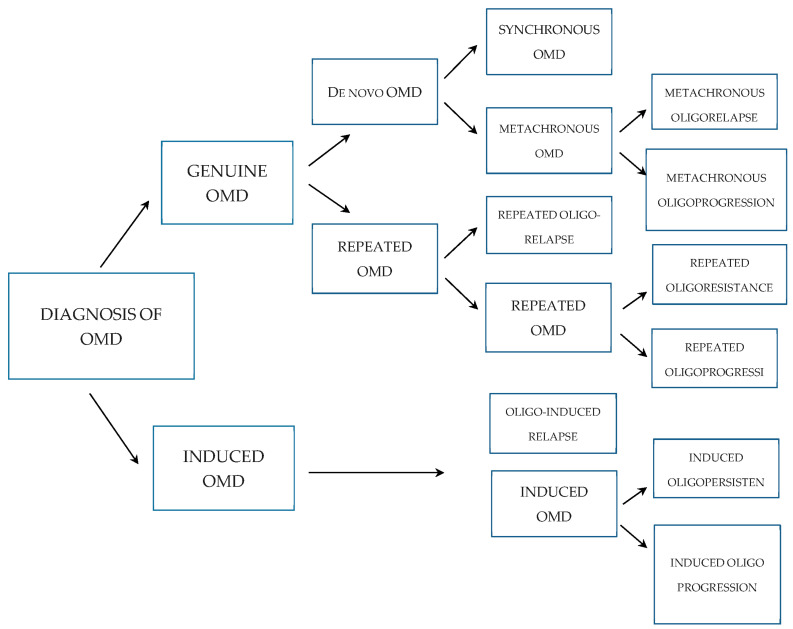
Classification of OMD.

**Figure 2 cancers-15-05234-f002:**
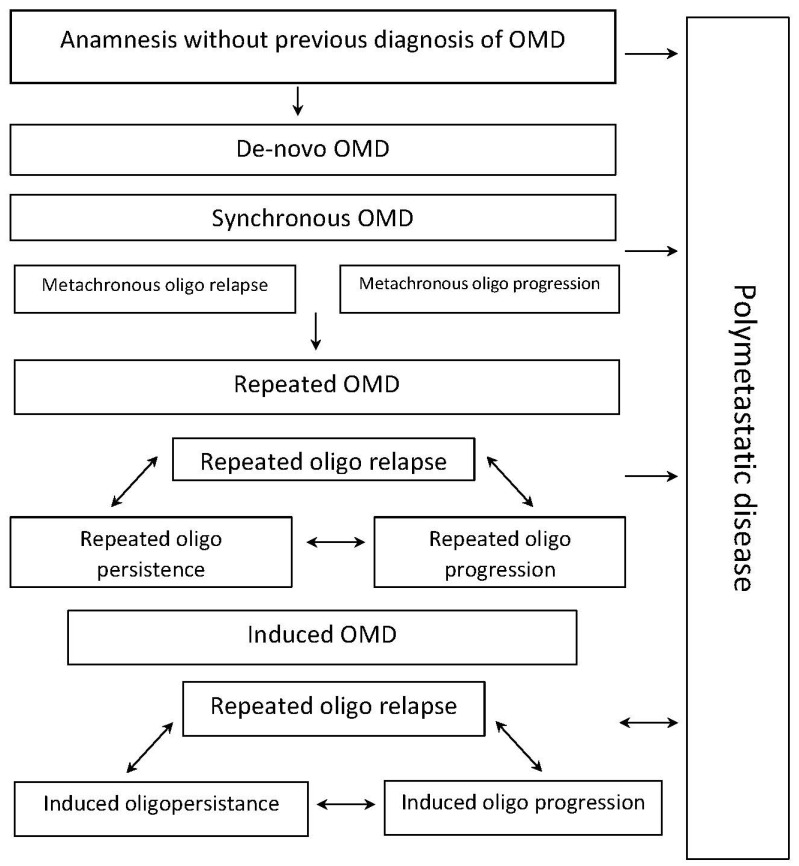
A dynamic model of OMD.

**Table 1 cancers-15-05234-t001:** Typical characteristics of OMD.

Data	Specifications
Characteristics of the tumor	Characteristics of the primary tumor: localization of the primary tumor, histology, TNM, mutation status, tumor marker. Progression data: time from the moment of the first diagnosis, disease-specific survival, no relapse period. Primary tumor treatment: local treatment method, radical or palliative treatment, status of the primary tumor. Systemic therapy before the diagnosis of OMD: types of systemic therapy, number of courses of drug therapy. Stage of OMD: imaging, extent of prevalence, invasion metastatic lesion in OMD.
Characteristics	Number of metastatic foci. Number of affected organs. Number of foci in the organ. Maximum size or volume of metastases.
Development characteristics	Did the patient have a history of gender and metastatic disease before the OMD diagnosis? Did the patient have a history of OMD before the current diagnosis? Was an OMD diagnosed within 6 months of the primary tumor diagnosis? Is the patient undergoing drug therapy at the time of OMD diagnosis? Progression?
Specific characteristics	Is an OMD lesion a newly developed metastatic lesion? Is radical treatment possible?

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
