# Peer review of "Oligometastatic Disease (OMD): The Classification and Practical Review of Prospective Trials"

_cancers, 2023, doi:10.3390/cancers15215234_

Round 1

Reviewer 1 Report

Comments and Suggestions for Authors

Title: A review of classification and trials of oligometastatic cancer introduction is a simplification. The international TNM classification includes three stages: T for local non-hematologic cancer, N for regional lymph node metastases, and M for distant metastases. Biologically, N is due to lymphogenic spread and M is due to hematogenous spread.  For colorectal cancer and breast cancer, the extent of N has prognostic significance and N is a tool for choice of adjuvant treatment. For cancers patients with N cancer has an overall survival between that for patients with T cancer and that for M cancer. The concept of oligometastatic cancer (OMC) is based on the fact that OMC can be treated with targeted treatment (surgery or radiation therapy) whereas oncologists treat patients with polymetastatic cancer (PMC) with systemic treatment. The implicit premise is that a metastasis-targeted treatment has the same impact on outcome whether the metastases are N or M metastasis.

Conventional imaging points to PMC and more sensitive second-generation imaging points to OMC. So imaging can contribute to OMC has a better survival than PMC.  l 79-82 the Ost phase II trial of patients with biochemically recurrent prostate cancer compared stereotactic body radiotherapy with surveillance. Endpoint was ADT-free survival, not overall survival.  The Ost paper did not report OS after ADT. The STAMPEDE trial examined impact of radiotherapy for the primary prostate cancer in patients with metastatic prostate cancer. The trial did not focus on OMC.

L 108. Second generation staging imaging modalities detect OMC, not X ray imaging of the chest.

L120 Relevance is impact of targeted radiation therapy for OMC NSCLC, not survival of patients with stage 1 NSCLC.

L159 Did the authors make a systematic review or report an effort for consensus, or both or neither. The review had no definition of OMC subgroups such as genuine, de novo, repeated, metachronous, oligoprogress, and repeated oligoprogression.

There are two phases of cancer: initial phase and phase after initial treatment (PSA relapse for prostate cancer).

1.Initial phase of cancer have three groups: T, OMC, and PMC.

2.Recurrence also have three groups T, OMC, and PMC.

2a.Initial T patients may recur as OMC.

2b.Initial OMC may persist, or regress and recur as OMC.

2c.Initial PMC may regress to T or less and later progress to OMC, or may regress to OMC.  How do the terms in the review fit to my five OMC subgroups.

L 178 Targeted treatment of OMC aims to prolong time to next salvage treatment, to prolong survival, and to cure.

L 193  Ref 11 and 12 evaluated local consolidative therapy (LCT) relative to maintenance systemic treatment or observation.

The review mainly reported published RCTs and did not report many ongoing RCTs of targeted treatments of OMC, for example for patients with prostate cancer. Limitation/Conclusion. No phase III RCT has proven that targeted treatment of OMC cures cancer.  

Comments on the Quality of English Language

Title: A review of classification and trials of oligometastatic cancer introduction is a simplification. The international TNM classification includes three stages: T for local non-hematologic cancer, N for regional lymph node metastases, and M for distant metastases. Biologically, N is due to lymphogenic spread and M is due to hematogenous spread.  For colorectal cancer and breast cancer, the extent of N has prognostic significance and N is a tool for choice of adjuvant treatment. For cancers patients with N cancer has an overall survival between that for patients with T cancer and that for M cancer. The concept of oligometastatic cancer (OMC) is based on the fact that OMC can be treated with targeted treatment (surgery or radiation therapy) whereas oncologists treat patients with polymetastatic cancer (PMC) with systemic treatment. The implicit premise is that a metastasis-targeted treatment has the same impact on outcome whether the metastases are N or M metastasis.

Conventional imaging points to PMC and more sensitive second-generation imaging points to OMC. So imaging can contribute to OMC has a better survival than PMC.  l 79-82 the Ost phase II trial of patients with biochemically recurrent prostate cancer compared stereotactic body radiotherapy with surveillance. Endpoint was ADT-free survival, not overall survival.  The Ost paper did not report OS after ADT. The STAMPEDE trial examined impact of radiotherapy for the primary prostate cancer in patients with metastatic prostate cancer. The trial did not focus on OMC.

L 108. Second generation staging imaging modalities detect OMC, not X ray imaging of the chest.

L120 Relevance is impact of targeted radiation therapy for OMC NSCLC, not survival of patients with stage 1 NSCLC.

L159 Did the authors make a systematic review or report an effort for consensus, or both or neither. The review had no definition of OMC subgroups such as genuine, de novo, repeated, metachronous, oligoprogress, and repeated oligoprogression.

There are two phases of cancer: initial phase and phase after initial treatment (PSA relapse for prostate cancer).

1.Initial phase of cancer have three groups: T, OMC, and PMC.

2.Recurrence also have three groups T, OMC, and PMC.

2a.Initial T patients may recur as OMC.

2b.Initial OMC may persist, or regress and recur as OMC.

2c.Initial PMC may regress to T or less and later progress to OMC, or may regress to OMC.  How do the terms in the review fit to my five OMC subgroups.

L 178 Targeted treatment of OMC aims to prolong time to next salvage treatment, to prolong survival, and to cure.

L 193  Ref 11 and 12 evaluated local consolidative therapy (LCT) relative to maintenance systemic treatment or observation.

The review mainly reported published RCTs and did not report many ongoing RCTs of targeted treatments of OMC, for example for patients with prostate cancer. Limitation/Conclusion. No phase III RCT has proven that targeted treatment of OMC cures cancer. 

Author Response

We thank very much the reviewer for the detailed analysis of our manuscript and comments and suggestions, as well. Below are our specific responses to the reviewer’s comments (shown in quotes and italics). The changes in the revised version of the manuscript are highlighted with yellow.

1.Conventional imaging points to PMC and more sensitive second-generation imaging points to OMC. So imaging can contribute to OMC has a better survival than PMC.  Yes, we totally agree with this point. The timely visualization of OMC contributes a better survival than for the patients  of polymetastatic disease.

  1. l 79-82 the Ost phase II trial of patients with biochemically recurrent prostate cancer compared stereotactic body radiotherapy with surveillance. Endpoint was ADT-free survival, not overall survival. The Ost paper did not report OS after ADT.” We greatly appreciate a reviewer for this comment and up-dated the revised version of our manuscript with the detailed information regarding this study. Indeed, the end-point of this study was ADT-free survival. The changes are highlighted with yellow (LL101-109 in the revised manuscript)
  2. “The STAMPEDE trial examined impact of radiotherapy for the primary prostate cancer in patients with metastatic prostate cancer. The trial did not focus on OMC”. We greatly appreciate the reviewer for this notice. Indeed, this trial was focused to evaluate the impact of radiotherapy for the primary metastatic PC. Even the authors did not focused on OMD state in PC, the patients with low metastatic burden exhibited the benefits of this therapeutic approach. All these points were described in the revised manuscript as shown in the LL 110-115.  
  3. L 108. Second generation staging imaging modalities detect OMC, not X ray imaging of the chest”. Yes, of course. To prevent this misunderstanding, we re-phrased these imaging modalities to “diagnostic imaging procedures including multispiral computer tomography” as shown in LL 133-134 in revised manuscript.
  4. L120 Relevance is impact of targeted radiation therapy for OMC NSCLC, not survival of patients with stage 1 NSCLC”. We discuss here the 5-year survival rates of the patients with non-small cell lung cancer depending from the stage of the disease (stages I-IV), including the overall survival rates in patients received the distant radiotherapy.
  5. L159 Did the authors make a systematic review or report an effort for consensus, or both or neither. The review had no definition of OMC subgroups such as genuine, de novo, repeated, metachronous, oligoprogress, and repeated oligoprogression”.

One of the aims of this review was analyze and interpret the benefits of the local ablative therapy for OMD state, illustrate the modern classification of OMD, and its dynamic state. All these point were also highlighted in the Figures 1 and 2. The definitions of OMC subgroups indicated above were also highlighted in the abstract of the manuscript.

  1. There are two phases of cancer: initial phase and phase after initial treatment (PSA relapse for prostate cancer).

1.Initial phase of cancer have three groups: T, OMC, and PMC.

2.Recurrence also have three groups T, OMC, and PMC.

2a.Initial T patients may recur as OMC.

2b.Initial OMC may persist, or regress and recur as OMC.

2c.Initial PMC may regress to T or less and later progress to OMC, or may regress to OMC.  How do the terms in the review fit to my five OMC subgroups”.  We agree with this point illustrating 2 cancer phases (the initial without metastatic leasons and recurrent one, which might be detected by biochemical analysis illustrating an increase of PSA levels in the serum or metastatic leasons that can be visualized by using imaging procedures.  Regarding the stages of PC, this is classification based on TNM, whereas T reflects the size of primary tumor, N - affects the regional lymph nodes and M represents the distant metastasis. Classification of OMC and PMC is described in details in Table 1 and Figure 1.

  1. L 178 Targeted treatment of OMC aims to prolong time to next salvage treatment, to prolong survival, and to cure”. We totally agree with this comment and highlighted this point in LL 212-214.
  2. L 193 Ref 11 and 12 evaluated local consolidative therapy (LCT) relative to maintenance systemic treatment or observation”. We agree with this notice and removed these references from this part of the manuscript.

Reviewer 2 Report

Comments and Suggestions for Authors Comments to the Author
The aim of the study is  to  review of the current concept of OMD and discuss the available prospective clinical trials and potential future directions
  I have a only a few comments: 1. ''randomized clinical trials (except one)'' Could the authors please explain what this one trial recommended? 2. figure 2: english characters are recommended  3. could the authors comment if there are evidence about the role of ablative techniques in oligometastatic disease?

Author Response

We thank the reviewer for the comments and suggestions regarding our manuscript. Below are our specific responses to the reviewer’s comments (shown in quotes and italics). The changes in the revised version of the manuscript are highlighted with yellow.

  1. ''randomized clinical trials (except one)'' Could the authors please explain what this one trial recommended?” This trial recommended to use the certain prognostic factors to further compare the results between the groups of patients and further evaluate this data.  
  2. figure 2: english characters are recommended”. We appreciate a reviewer for this notice and fixed the typos shown on the bottom of the figure 2.
  3. “could the authors comment if there are evidence about the role of ablative techniques in oligometastatic disease?” We appreciate the reviewer for this question. For example, the manuscript entitled “Ablative Techniques for Sarcoma Metastatic Disease: Current Role and Clinical Applications” was published recently and describes the evidence about the role of ablative techniques in OMD. (Efthymiou E, Charalampopoulos G, Velonakis G, Grigoriadis S, Kelekis A, Kelekis N, Filippiadis D. Medicina (Kaunas). 2023 Mar 1;59(3):485. doi: 10.3390/medicina59030485. PMID: 36984486).

Round 2

Reviewer 2 Report

Comments and Suggestions for Authors

Dear authors,

all queries addressed according to reviewer’s comments.